# Mass and Heat Transfer of Thermochemical Fluids in a Fractured Porous Medium

**DOI:** 10.3390/molecules25184179

**Published:** 2020-09-12

**Authors:** Murtada Saleh Aljawad, Mohamed Mahmoud, Sidqi A Abu-Khamsin

**Affiliations:** College of Petroleum Engineering & Geosciences, King Fahd University of Petroleum & Minerals, Academic Belt Road, Dhahran 31261, Saudi Arabia; mmahmoud@kfupm.edu.sa (M.M.); skhamsin@kfupm.edu.sa (S.A.A.-K.)

**Keywords:** mass transfer, heat transfer, thermochemical fluids, modeling, limestone fracture

## Abstract

The desire to improve hydraulic fracture complexity has encouraged the use of thermochemical additives with fracturing fluids. These chemicals generate tremendous heat and pressure pulses upon reaction. This study developed a model of thermochemical fluids’ advection-reactive transport in hydraulic fractures to better understand thermochemical fluids’ penetration length and heat propagation distance along the fracture and into the surrounding porous media. These results will help optimize the design of this type of treatment. The model consists of an integrated wellbore, fracture, and reservoir mass and heat transfer models. The wellbore model estimated the fracture fluid temperature at the subsurface injection interval. The integrated model showed that in most cases the thermochemical fluids were consumed within a short distance from the wellbore. However, the heat of reaction propagated a much deeper distance along the hydraulic fracture. In most scenarios, the thermochemical fluids were consumed within 15 ft from the fracture inlet. Among other design parameters, the thermochemical fluid concentration is the most significant in controlling the penetration length, temperature, and pressure response. The model showed that a temperature increase from 280 to 600 °F is possible by increasing the thermochemical concentration. Additionally, acid can be used to trigger the reaction but results in a shorter penetration length and higher temperature response.

## 1. Introduction

Recently, thermochemical fluids (TCs) have been introduced in various applications in the oil and gas industry. Thermochemicals are fluids whose reaction generates high temperatures and pressures. The amount of heat and pressure generated is a function of the concentration of the thermochemicals and initial conditions of the reaction medium. The reaction can be triggered by acid or when reaching a certain temperature

TCs are used to generate very-high-pressure pulses in order to fracture rocks. The pressure pulses generated by to the chemical reaction can reach more than 10,000 psi, creating complex fracture networks [1]. This method is applied to increase the stimulated reservoir volume (SRV) around hydraulic fractures in unconventional and tight reservoirs. Hotspots (i.e., regions with multiple fractures of various sizes) are often observed after TC treatment. The pore structures of the rocks are changed due to the pressure and temperature generated [2,3]. On a large scale, TCs have proven effective in enhancing the SRV [4]. Hydraulic fracturing in layered reservoirs can be controlled via TC pulse fracturing to initiate a fracture in different layers and prevent inter-layer fracturing. This technique enhances fracture efficiency (i.e., complexity) and extension [5].

TCs have often been implemented to ease production in heavy oil reservoirs. They tend to be more efficient than steam injection. The advantage of TC use is that heat and pressure are generated in situ; no heat loss is reported, in contrast to steam injection. TC applications in heavy oil recovery offer a particular advantage in deep wells and cold areas in which steam might create wellbore integrity issues. Different techniques and injection modes have been suggested to enhance the efficiency of thermochemicals in heavy oil recovery from different types of reservoirs [6,7,8,9,10].

Fracturing fluids usually contain crosslinked fluids in order to create wide fractures and improve proppant suspension. TCs are used to break gels and high-viscosity polymers during fracturing to clean up the well and remove polymer residue, thus enhancing well deliverability and reducing the fracture’s face skin [11]. Hassan et al. [12,13] used TCs to remove damage attributable to condensate banking and water blockage in tight gas reservoirs. The authors have proved that thermochemical treatment has better performance in terms of removing condensate than existing methods. TCs change the thermodynamics and phase behaviors of the condensate liquid in the near-wellbore area and turn the phase to gas. Large-scale simulation and experimental studies on tight and unconventional rocks show more than a 60% increase in gas production after thermochemical treatment. TCs desiccate the water in the near-wellbore area and the nitrogen and heat generated removed water blockages caused by drilling fluid and mud filtrate. In addition, the thermochemical reaction generates pressure pulses that created micro-fractures in tight rocks. These micro-fractures enhanced the overall permeability by three orders of magnitude. In turn, condensate and water blockage formation is eliminated due to the reduction in reservoir drawdown.

TCs have been implemented in various field trials to minimize damages in different kinds of wells and restore well productivity/injectivity [2]. Tar residue can cause severe damage in water injection and oil production wells. Alade et al. [14] used thermochemical treatments to remove tar damage, showing that pressure and heat generated due to thermochemical reactions improved tar emulsification and mobilized tar to the surface at a low viscosity. In addition, the authors used the same concept to remove damage due to various organic scales, such as asphaltene and emulsion blockage, in different conditions [15]. Wax deposition was eliminated by the heat generated from the reaction and continuous injection of the thermochemicals at different cycles.

Stimulation of sandstone and carbonate reservoirs using thermochemicals showed very good results in both the laboratory and the field. Hassan et al. [16] demonstrated that thermochemical treatments enhanced the permeability of a tight sandstone outcrop sample from 0.2 to 300 mD. The authors attributed the significant improvement in permeability to the creation of a dominant fracture throughout the entire sample. They also verified the effect of overburden pressure on permeability decline, finding a constant permeability value across a wide range of overburden pressures. This was due to the creation of etched surfaces during thermochemical fracturing. Gomaa et al. [17,18] found similar results in terms of permeability improvement in different sandstone rocks with a variety of clay contents.

The thermochemicals targeted in this study are those that when reacting produce heat, pressure, and follow the stoichiometric equation below [19]:(1)A+B→C  ΔH <0

The enthalpy (ΔH) depends on the type and state of the reactants, whether liquid or solid. In this case, the reaction is exothermic, and the enthalpy has a negative value. Different chemicals can be used to generate the heat and pressure required. In this research, the reaction between ammonium chloride (${\mathrm{NH}}_{4}\mathrm{Cl}$) and sodium nitrite (${\mathrm{NaNO}}_{2}$) was studied.

Mass and heat transfer models of a system containing wellbore, hydraulic fracture, and reservoir have been developed previously [20,21,22]. Even though TC reactions have been analyzed extensively in lab experiments and applied frequently in petroleum engineering applications, a field-scale model has yet to be developed. Such a model will help explain the impact of TCs on reservoir conditions. For instance, the results of the present research will provide information about the penetration length and heat propagation of TCs within a reservoir. Design conditions, such as the injection rate, treatment volume, and molar concentration, can then be determined, based on these results.

## 2. Modeling Methodology

An integrated mass and heat transfer model were built to describe the reactive transport of TCs in a hydraulic fracture. The model was developed in-house using MATLAB R2019a. The modeled system contains subterranean tight formation, a wellbore drilled in the middle of the formation, and a fracture extending from the wellbore. The model is dynamic, in that the fracture propagates while the fluids containing TCs are being injected. TCs generate an instantaneously high pressure and temperature, which are believed to improve fracture complexity. The reaction can be triggered by acid or when reaching a certain temperature. The model assumes that the reaction takes place once the fluids are inside the formation. The goal of the present work was to understand the penetration length of TCs and their heat propagation distance within hydraulic fractures. Modeling the complex fracture networks caused by TCs was not the focus of this study.

Figure 1 illustrates the mathematical sequence of the model. Fluid mass and heat transfer are simulated in the wellbore, fracture, and formation. At each time step, the mass and heat transfer in the wellbore and surrounding formation are solved. Fluids are injected from the wellbore into the formation, creating the hydraulic fracture. The fracture dimensions and velocity profile are then estimated. Notice that the flow in the fracture induces the flow in the reservoir, due to fluid leak off. The TC mass and heat transfer inside the fracture are then obtained. This information is coupled with the mass and heat transfer in the reservoir until the final injection time is reached. Simulations are also conducted during shut-in in order to estimate the wellbore temperature after stimulation. Notice that the model can also simulate acid reaction in the fracture, which can be used to trigger the TC reaction. The different colored boxes in the flow diagram show the various models used in this study.

### 2.1. Velocity Profiles

In the petroleum industry, modeling of the reactive transport of acid systems inside hydraulic fractures has been researched extensively [23,24,25,26,27]. Similar concepts were used to model TCs in this research, but the appropriate reaction kinetics and boundary conditions (BCs) were applied [28,29]. The units of parameters in both SI and field metrics are shown in the nomenclature.

The total and component mass transfers can be applied to a system containing a wellbore, hydraulic fracture, and formation in order to obtain both the velocity and chemical concentration profiles. Since fluid injection begins at the wellbore surface, the velocity profile in the wellbore should be obtained first. The one-dimensional (1-D) continuity equation in the wellbore can be written as:(2)$$\frac{\partial \rho_{f}}{\partial t}\;=\;\frac{2\gamma }{R_{w}}\rho_{f}u_{L,p}-\frac{\partial \left(\rho_{f}u_{z}\right)}{\partial z}$$
where $\rho_{f}$ (kg/m^3^) is the fracture fluid density, γ is the wellbore-open fraction (at the injection interval), t (s) is the time, $R_{w}$ (m) is the radius of the wellbore, $u_{L,p}$ (m/s) is the velocity of fluids entering the formation, and z (m) is the distance along the wellbore’s length. The first term in the above equation is the mass accumulation, the second term is the mass exiting the wellbore, and the final term is the mass advection within the wellbore. The equation is used to obtain the velocity profile in the wellbore. It assumes single-phase fluid and the impact of heat transfer on the velocity profile is negligible. Figure 2 shows a schematic of the domain containing the wellbore. The fluids are injected to the wellbore where they travel down the hole until reaching the wellbore perforations. It creates cracks where the pressurized fluids propagate the hydraulic fracture. The red arrows show the directions of the fluid flow in the domain.

Then, the fluid velocity distribution inside the hydraulic fracture can be obtained. The solution should consider the fluid loss’s impact on fluid convection within the hydraulic fracture. Figure 3 shows the two-dimensional (2-D) schematic of the domain by taking a slice in the z-direction (top view). The continuity and momentum balance equations are utilized only in the hydraulic fracture, and are solved based on the Berman [30] approach:(3)∇·u=0
(4)$$\rho_{f}\left(u\cdot \nabla u\right)=-\nabla p+\mu \Delta u$$
where u (m/s) is the velocity vector, μ (kg/m·s) is the fracture fluid viscosity, and p (pa) is the pressure inside the fracture. The continuity and momentum balance in the fracture are applied in 2-D (see Figure 4) and assumes the following:(1)Incompressible single-phase fluids;(2)Newtonian fluids while the effective viscosity is used for non-Newtonian fluids;(3)Steady-state condition;(4)Laminar flow; and(5)Leaky channels where fluids are lost from the fracture to the reservoir.

The fracture inlet boundary condition (BC) is the velocity of the fluids leaving the wellbore, $u_{L,p}$. Fluid leaking from the fracture will induce flow in the reservoir. The fluid loss from the fracture can be estimated using Carter’s equation, as follows (Equation (5)):(5)$$u_{L}=\frac{C_{L}}{\sqrt{t-\tau \left(x\right)}}$$
where $C_{L}$ (m/s^1/2^) is the fluid leakoff (loss) coefficient and τ(x) (s) is the time when the fracture propagates to position x. Equation (5) is used as the wall outlet BC for the fracture (see Figure 4) and inlet BC for the reservoir domain (see Figure 3). The flow in the porous media surrounding the fracture can be obtained by solving the diffusivity equation:(6)$$\nabla \cdot \left(k\cdot \nabla p\right)=\varphi \mu_{r}c_{t}\frac{\partial p}{\partial t}$$
where p (pa) is the reservoir pressure, φ is the rock porosity, k (m^2^) is the permeability tensor,$\;c_{t}$ (1/pa) is the total compressibility, and $\mu_{r}$ (kg/m·s) is the reservoir fluid viscosity. The diffusivity equation is a combination between Darcy’s law and the continuity equation in a porous medium. The diffusivity equation assumes incompressible, single-phase, and laminar flow. Notice that the simulations are applied only to a quarter of the reservoir domain, due to symmetry. This significantly reduces the computational cost of the model. Figure 3 shows the different BCs applied to solve the equation. The pressure distribution (obtained from the diffusivity equation) can be converted to a velocity profile through Darcy’s law, as follows:(7)$$u=-\frac{1}{\mu_{r}}\left(k\cdot \nabla p\right)$$

### 2.2. Concentration Profiles

After obtaining the velocity distribution inside the hydraulic fracture and reservoir, chemical concentrations and temperature profiles can be determined. Hydrochloric acid (HCl) can be used to trigger the TCs’ reaction and create etching along the fracture. The model assumes that acid reacts with the carbonate rock minerals (a heterogeneous reaction), while the TCs react with each other (a homogenous reaction). The chemical reactions can be explained as follows for both the HCl acid with limestone (Equation (8)) and TCs (Equation (9)):(8)$$2HCl+CaCO_{3}\to CaCl_{2}+CO_{2}+H_{2}O+\Delta H$$
(9)$$NH_{4}Cl+NaNO_{2}\to NaCl+2H_{2}O\;+N_{2}+\Delta H$$

To track the concentration of each component (i.e., the reactants) along the fracture, the component mass balance must be solved. For the TCs, the following equation is applied:(10)$$\left(\frac{\partial C_{TC}}{\partial t}+u\cdot \nabla C_{TC}\right)\;=\;\nabla \cdot \left(D_{TH}\nabla C_{TC}\right)+r_{TC}$$
where $C_{TC}$ (mol/m^3^) is the reactant concentration (${\mathrm{NaNO}}_{2}$), $D_{TH}$ (m^2^/s) is the reactant diffusion coefficient, and $r_{TC}$ (mol/m^3^·s) is the reaction rate of the thermochemical fluids. The first term represents the TC accumulation, the second term is the TC convection, the third term is the diffusion, and the final term is the reaction. The diffusion term can be ignored as it has no noticeable impact on the solution. The TC reaction rate is first order according to Alade et al. [31], and follows the Equation (11) below:(11)$$r_{TC}=\;k_{r}C_{TC}$$
where $k_{r}$ (1/s) is the reaction rate constant. In this model, it is assumed that there is no TC concentration gradient, ${\frac{\partial C_{TC}}{\partial y}|}_{w}=0$, at the fracture walls (i.e., the north and south boundaries). It is also assumed that the reaction takes place inside the fracture (i.e., there is no reaction in the wellbore), where the inlet concentration is the initial TC concentration, ${C_{TC}|}_{inlet}=\;C_{i}$ (see Figure 4).

The reaction between the HCl and rock minerals occurs at the fracture walls (i.e., the south and north domain boundaries). The acid mass balance can be written as:(12)$$\left(\frac{\partial C_{A}}{\partial t}+u\cdot \nabla C_{A}\right)=\;\nabla \cdot \left(D_{A}\nabla C_{A}\right)$$
where the subscript *A* stands for acid. The first term of the equation above is the acid accumulation, the second term is the acid advection, and the last term is the acid diffusion. Notice that the reaction does not take place in bulk, as was the case for the TCs. The reaction appears as BCs at the fracture walls (see Figure 4), according to the following equation:(13)$$-D_{A,y}{\frac{\partial C_{A}}{\partial y}|}_{w}=r$$
where *r* (mol/m^2^·s) is the reaction rate of the acid. The boundary states that the rate of acid diffusion to the fracture walls is equal to the reaction rate. Acid/rock reaction kinetics can be expressed as:(14)$$r=k_{r}C_{A,w}^{n_{r}}\left(1-\varphi \right)$$
where $k_{r}$ ($\frac{kg\;moles\;HCl}{m^{2}\cdot s\cdot \left(\frac{kg\;moles\;HCl}{m^{3}\;acid\;solution}\right){\;}^{n_{r}}}$) is the reaction rate constant and $n_{r}$ is the reaction exponent. The acid dissolves the fracture walls according to the following:(15)$$\frac{\partial w_{e}}{\partial t}=\frac{X}{1-\varphi }\left(f_{a}u_{L}C_{A,w}-{D_{A}\frac{\partial C_{A}}{\partial y}|}_{w}\right)$$
where $w_{e}$ (m) is the etched width by acid, $f_{a}$ is the fraction acid lost that reacts at the fracture surfaces, χ is the volumetric dissolving power, and $u_{L}$ (m/s) is the leakoff velocity. The equation states that the change in fracture-etched width is due to acid convection (second term) and diffusion (third term) towards the fracture walls.

### 2.3. Temperature Profiles

When fracture fluids are injected from a wellbore surface, they are heated by the surrounding warm formation. Knowing the temperature of the TCs before they enter the formation is necessary for obtaining accurate results regarding TC penetration. Hence, it is essential to solve the energy balance in the wellbore, which is written as:(16)$$\rho_{f}\hat{C}p_{f}\left(\frac{\partial T_{wb}}{\partial t}+u_{z}\frac{\partial T_{wb}}{\partial z}+\frac{2\gamma }{R}u_{L,p}\left(T_{wb}-T_{f|B}\right)\right)=\frac{2\left(1-\gamma \right)U}{R_{w}}\left(T_{r|B}-T_{wb}\right)$$
where $T_{wb}$ (°C) is the temperature along the wellbore, $T_{f|B}$(°C) is the fracture temperature at the wellbore/fracture contact,$\;\hat{C}p_{f}$ (kJ/Kg °C) is the specific heat capacity of the fluids, U (kJ/s·m^2^ °C) is the overall heat transfer coefficient, and $T_{r|B}$ (°C) is the temperature at the wellbore/formation periphery. Figure 2 shows a schematic of the wellbore within the domain. The first term in the equation above represents the heat accumulation, the second term is the heat convection along the wellbore length, the third term is the heat convection from the fluids lost to the fracture, and the final term is the heat conducted from the formation. The equation is one-dimensional (1-D), and essentially ignores the heat generation due to friction. The model assumes that the thermal properties, such as $\hat{C}p_{f}$ and U, are constant and do not change with location or time. It is also assumed that the reaction does not take place in the wellbore. Solving this equation requires knowing the fracture and formation temperatures. The formation temperature surrounding the wellbore can be obtained by solving the heat conduction equation:(17)$$\bar{\rho \hat{C}p}\frac{\partial T_{r}}{\partial t}=\nabla \cdot \left(\bar{k_{e}}\nabla T_{r}\right)$$
where $\bar{\rho \hat{C}p}$ represents the average rock and fluid properties and $\bar{k_{e}}$ (kJ/s·m °C) is the average rock and fluid thermal conductivities. The equation states that the temperature change in the formation surrounding the wellbore is due to heat conduction. The formation temperature model assumes constant thermal conductivities. It also assumes that there are no fluid communications between the wellbore and the formation except through the perforated zone. Notice that the wellbore and formation temperature are coupled through the shared boundary (see Figure 2).

Heat transfer in the fracture and reservoir takes place because of the difference between the temperatures of the injected fluids and reservoir. Moreover, the injected fluids release heat due to the exothermic reaction, significantly altering the temperature profile. Obtaining the temperature profile is very important for estimating the concentrations of reactive components, and vice versa. This is because the reaction rate constant depends on the temperature, according to the Arrhenius equation (Equation (18)):(18)$$k_{r}=k_{r}^{0}exp\left(-\frac{\Delta E}{RT}\right)$$
where $k_{r}^{0}$ ($\frac{kg\;moles\;HCl}{m^{2}\cdot s\cdot \left(\frac{kg\;moles\;HCl}{m^{3}\;acid\;solution}\right){\;}^{n_{r}}}$) is the pre-exponential factor, ΔE (kJ/mol) is the activation energy, R (m^3^×Pa/K×mol) is the universal gas constant, and T is the absolute temperature. Notice that both the thermochemical fluids and acid reaction rates depend strongly on temperature according to Equation (18) but has no dependence on pressure. Similarly, the diffusion of fluids is a function of the temperature, according to the Arrhenius formula:(19)$$D=D_{o}exp\left(-\frac{\Delta E_{D}}{RT}\right)$$
where $\Delta E_{D}$ (kJ/mol) is the activation energy for the diffusion. The heat transfer model in the fracture can be written as:(20)$$\rho_{f}\hat{C}p_{f}\left(\frac{\partial T_{f}}{\partial t}+u\cdot \nabla T_{f}\right)=\;\nabla \cdot \left(k_{f}\nabla T_{f}\right)+R\;\Delta H_{r,TC}$$
where $T_{f}$ (°C) is the fracture temperature, $\hat{C}p_{f}$ (kJ/kg °C) is the fluid specific heat capacity, $k_{f}$ (kJ/s·m °C) is the fluid thermal conductivity, and $\Delta H_{r,TC}$ (kJ/mol) is the thermochemical heat of reaction. Notice that the first term represents the heat accumulation, the second term is the heat convection, the third term is the heat conduction, and the final term is the heat of reaction released within the fracture because of the TCs’ reaction. The HCl acid reaction with the minerals releases heat as well but appears as a BC as follows (see Figure 5):(21)$${-k_{f,y}\frac{\partial T_{f}}{\partial y}|}_{w}=\left|r\left(\Delta H_{r,A}\right)\right|+q_{r}$$
where $\Delta H_{r,A}$ (kJ/mol) is the acid heat of reaction, and $q_{r}$ (kJ/s·m^2^) is the heat lost to the reservoir. The fracture heat transfer model is 2-D as shown in Figure 5. All thermal properties are assumed to be constant. The boundary states that the heat conducted towards the fracture wall is equal to the heat of reaction and heat lost to the formation. The heat loss from the reservoir can be described as:(22)$$q_{r}=\bar{k_{e}}{\frac{\partial T_{r}}{\partial y}|}_{w}$$

The heated fracture fluids transfer heat to the reservoir. The heat transfer in the reservoir can be described through the heat convection and conduction equation (Equation (23)):(23)$$\bar{\rho \hat{C}p}\frac{\partial T_{r}}{\partial t}+\rho_{f}\hat{C}p_{f}u\cdot \nabla T_{r}=\nabla \cdot \left(\bar{k_{e}}\nabla T_{r}\right)$$
where $\bar{\rho \hat{C}p}$ is the average rock and fluid properties. The first term is the heat accumulation, the second term is the heat convection within the reservoir due to fluid leakoff from the fracture, and the final term is the heat conduction. This equation is applied to the reservoir section perpendicular to the fracture (see Figure 5). Similar equations are applied to simulate heat transfer during shut-in by omitting the convection terms in the above equations.

### 2.4. Fracture Domain

The fracture dimensions should be estimated at each time step. The fracture geometry represents the fracture length, width, and height. This is done by applying the following 1-D material balance equation to the fracture domain:(24)$$\frac{\partial q}{\partial x}+2u_{L}h_{f}+\frac{\partial A_{c}}{\partial t}=0$$
where q (m^3^/s) is the flow rate inside the fracture, $u_{L}$ (m/s) is the fluid loss velocity from the fracture to the reservoir, $h_{f}$ (m) is the fracture height. $A_{c}$ (m^2^) is the fracture cross-section area, x is the position along the fracture length, and t (s) is the injection time. The equation states that the fluid flowing in the fracture is either lost to the formation or stored in the fracture. The equation above is used to determine the fracture length assuming the width is known. To estimate the cross-sectional area in Equation (23), the fracture width must be estimated. The dynamic fracture width is calculated according to the Perkin-Kern-Nordgren (PKN) model assumptions as follows:(25)$$w_{max,0}=9.15^{\frac{1}{2n+2}}\;\;3.98^{\frac{n}{2n+2}}{\left(\frac{1+2.14n}{n}\right)}^{\frac{n}{2n+2}}K^{\frac{1}{2n+2}}{\left(\frac{q_{i}{}^{n}h_{f}^{1-n}x_{f}}{È}\right)}^{\frac{1}{2n+2}}$$
where $w_{max,0}$ (m) is the maximum fracture width, $x_{f}$ (m) is the fracture half-length, n is the power law exponent, K is the consistency index, $q_{i}$ (m^3^/s) is the injection rate, and È (pa) is the plain strain modulus. For simplicity, in the present study, the fracture height was assumed to be constant. The fracture model is a simplified 2-D that assumes plane strain. It is assumed that as the fluid is being injected and it is larger than the fluid loss, the fracture propagates. Table 1 and Table 2 show the data used for the simulations employed in this study. The data in Table 1 are based on typical wellbore, reservoir, and fluid properties that are encountered in carbonate reservoirs. The data in Table 2 are based on lab reaction kinetics measurements. The activation energy for the acid diffusion was 12.72 KJ/mol, according to Roberts and Guin [32].

### 2.5. Model Limitations

Modeling TCs’ reactive transport is a complex phenomenon as the reaction results in multiphase flow and possible turbulence. The developed model assumes that the TCs are incompressible (even after reactions) and the flow regime to be laminar. The component mass transport accounts for the reactants concentration but ignores the products of the reaction, such as CO_2_ and N_2_. The complex fracture network created by the TCs reaction is also ignored and planar fracture is assumed. Additionally, the thermochemical fluids were assumed to be well mixed while flowing in the fracture which is reasonable if they were co-injected at the wellbore perforations.

### 2.6. Solution Grid Independence

This section shows the optimum grid size selection for the numerical solutions. This was done by refining the grid blocks in both the fracture length (NX) and fracture width (NY) directions and observing the solution error reduction as compared to the most refined solution (NX = 18,750 and NY = 6250). Figure 6 shows the solutions in dimensionless form at different grid sizes. Figure 6a shows that when NX = 150, the solution becomes almost similar to the most refined solution and results in small root mean square error (RMSE = 1 × 10^−3^). Figure 6b shows that when NY = 50, the results become similar to the most refined solution with RMSE = 10^−4^. In this study, NX = 700 and NY = 80 were selected to lower the solution error without resulting in high computational cost.

## 3. Model Validation

To build confidence, the developed model was verified against lab experimental data. Verification against field data was not pursued due to its scarcity. Al-Nakhli et al. tested the ability of thermochemical fluids on generating high-temperature and -pressure pluses in an autoclave reactor [34]. The reactor is manufactured to handle a high-pressure and high-temperature (HPHT) environment. The reaction was monitored remotely using a personal computer (PC) for safety purposes where the temperature and pressure were recorded every 2 s. These reactors are usually made of high-quality jacket that is made of stainless steel and inner Teflon champers. Al-Nakhli et al. specified the different used reactor volumes and their pressure and temperature ratings in detail [34]. Figure 7 shows the test outcomes, where the thermochemical fluids could raise the temperature up to 600 °F and pressure up to 3300 psi. The reaction took place at 50 min, where a sudden increase in pressure and temperature was observed. To match the lab data, the thermochemical reaction parameters in Table 2 were used. To obtain a good temperature match (see Figure 7a), the overall heat transfer coefficient (U) was assumed to be changing between 0.01 and 0.4 KJ/(s·m^2^ °C) during the reaction. For the pressure match (see Figure 7b), the total compressibility was tuned to be around 1 × 10^−6^ 1/psi. The model root mean square error (RMSE) for temperature match was 30 °F while for the pressure match it was 120 psi.

## 4. Model Results and Analysis

This section discusses the TC penetration distance during a hydraulic fracture treatment. Heat propagation through the fracture and reservoir are also described. The TC penetration distance is defined as the location inside the fracture where the TCs concentration is zero. Other studies prefer to use a percentage of the initial chemical concentration for penetration distance calculations. Thermochemical reaction usually results in a sharp concentration front, which makes it indifferent to using a zero concentration or another slightly higher concentration. The heat propagation distance is the location in the fracture where the fluids’ temperature drops to the initial reservoir temperature. In this study, the reaction was assumed to be triggered at 102 °F. It should be noted, however, that the triggering temperature could be altered according to the specification of a certain field. The first subsection includes an analysis of the TC reaction phenomena occurring during injection. The second subsection analyzes a sensitivity analysis of the impacts of different design conditions (e.g., injection rate, TC concentration, fluid injection temperature). The third subsection discusses the outcome when the TC reaction was triggered by HCl acid. Additionally, the dissolution profiles from when the TCs were mixed with HCl acid are compared to the acid injection-only case. All the results are presented for the last time step unless stated otherwise.

### 4.1. Thermochemical Reaction

This section discusses a case study of 600 bbl of fracture fluids injected at 20 bpm, where the TC concentration was 3 ${\mathrm{NH}}_{4}\mathrm{Cl}$ mol/L to 3 ${\mathrm{NaNO}}_{2}$ mol/L (3:3 mol/L). It was assumed that the fluids were injected at 95 °F from the surface and traveled 8000 ft within the wellbore before being injected in the reservoir. The fluids were heated within the wellbore (due to heat flux from the formation) before reaching the injection point. The model assumed that the reaction occurred only once the fluids entered the formation. Nevertheless, the reaction could have taken place in the wellbore, a scenario to be avoided. At a steady state, the fluids entered the formation at 102 °F. Figure 8 shows the TC 2D concentration profile along the fracture half-length and width after 30 min of injection (i.e., the end of treatment). The fracture inlet was the western part of the domain, where the inlet TC concertation was 3:3 mol/L. In this case, the TCs penetrated to around 15 ft inside the fracture before total consumption. It should be noted that the hydraulic fracture propagated to a much longer distance (230 ft). It should be noted that the continuous injection of fracture fluids at a high rate is what caused the fracture propagation. Figure 9a shows a 1-D concentration profile at a very early injection time. During that time, the TCs propagated quickly, as the reservoir temperature was relatively low. Then, the heat generated from the TCs increased the temperature to higher levels, causing the reaction to be faster and the penetration distance to decrease (see Figure 9b). Once the temperature magnitude near the wellbore reached an approximately constant value, the TC penetration distance increased again at a steady pace, as shown in Figure 9c. Notice that the early time figures were in seconds while the late time was in minutes. Figure 10 shows the temperature profile along the fracture length at different injection times. Notice that the reservoir’s initial temperature was assumed to be 212 °F, which is a typical value for many reservoirs. Even though the TCs were consumed within 15 ft of the wellbore, the heat of reaction propagated a much greater distance, as the figure indicates. The maximum temperature magnitude occurred a couple of feet away from the fracture inlet where the reaction was the fastest. Then, it dropped along the fracture half-length and reached the initial reservoir temperature at 180 ft, as the last treatment time shows. The initial sharp increase in temperature is caused by the fast exothermic reaction, which was assumed to be initiated in the reservoir. The temperature peak is associated with the location at which the thermochemical fluids are completely consumed. Then, the temperature decreases due to the heat loss to the colder reservoir, as fluids are flowing in the fracture. At a certain point, the fracture fluids’ temperature reaches a plateau, which is the initial reservoir temperature, as illustrated in Figure 10. Several fundamental studies concluded that the temperature peak occurs away from the wellbore in reactive transport problems [35,36,37] The results are significant, showing that the heat propagated a long distance within the fracture even though the TCs were consumed within a short distance. In this case, a 400 °F increase in the temperature was realized at a certain point inside the fracture.

The heat could also have propagated through the reservoir matrix due to fluid lost from the fracture and heat conduction. Figure 11 shows a portion of the reservoir quarter domain where the well was placed at the origin and the fracture in the southern part. During injection, the heat propagated mostly along the fracture length and could reach only a small distance within the formation orthogonal to the fracture during the 30 min of injection (see Figure 11a). However, the heat propagated to a deeper distance after 7 h of shut-in, as shown in Figure 11b. The heat could have reached to a couple of feet within the formation depth orthogonal to the fracture after several days of shut-in.

### 4.2. Sensitivity Analysis

A change in the design conditions could significantly alter the results obtained above. Among these conditions are the fracture fluid temperature, fluid injection rate, molar concentration of the TCs, and treatment volume. The data from Table 1 and Table 2 were used for the simulations described below.

#### 4.2.1. Fluid Injection Temperature

It was assumed in this model that relatively cold fracture fluids were injected from the surface into a warmer formation. If no reaction takes place in the wellbore, the fluids enter the formation at a much lower temperature than the reservoir’s, as shown in the study above. Nevertheless, a premature reaction may take place in the wellbore that causes the fluid temperature to be higher before reaching the formation. A higher fluid injection temperature could be also due to the higher surface injection temperature or longer distance fluids need to travel within the wellbore. For instance, if the fracture fluids enter the formation at a temperature similar to that of the reservoir (212 °F), the TCs will penetrate a much shorter distance (2 ft), as shown in Figure 12a, due to the increase in the reaction rate. This is a much shorter penetration distance than the 15 ft obtained above. A higher fracture temperature profile was realized in this case, where the temperature peak was closer to the wellbore. Simulations considering different fracture fluid temperatures are shown in Figure 13a. It can be observed that the higher the injection temperature, the lower the TC penetration distance. Additionally, the higher the injection temperature, the higher the temperature generated; this is due to the exothermic reaction (see Figure 13b). The reaction takes place along a shorter distance, causing more energy per unit volume to be released. Moreover, the higher the injection temperature, the closer the reaction peak to the wellbore, as shown in Figure 13b. The TC fluids reaction is a strong function of temperature and hence the reaction occurred faster, resulting in the peak being closer to the wellbore. The peak does not occur exactly in the wellbore due to the assumption that the reaction takes place inside the formation. In all cases, however, the heat propagation distances were found to be similar (look at the plateau). This is because the fracture size and fluids advection within the fracture were similar, resulting in similar heat loss characteristics. Notice that the areas under the curves in Figure 13b were not similar because the fluids were assumed to have different energy levels (temperatures) before entering the formation.

#### 4.2.2. Fluid Injection Rate

Fluids can be injected at various rates, depending on the purpose of the stimulation job. If near-wellbore cleaning is required, fluids are usually injected at low rates. However, if a hydraulic fracture is the intent, higher injection rates are targeted. This research focused on understanding the reactive transport of TCs in hydraulic fractures. Figure 14 shows the simulation results from different injection rates, representing the typical range of hydraulic fracture design. It was determined that the higher the injection rate, the greater the TC penetration distance (see Figure 14a). This is logical because the TCs travel at a higher velocity, reducing their residence time. The injection rate can also significantly impact the temperature profile. Figure 14b shows that the higher the injection rate, the greater the heat propagation distance. This is because higher injection rates result in longer fractures. Additionally, hot fluids travel long distances within the fracture before being cooled by the reservoir due to higher fluid velocity. Thus, increasing the injection rate results in the temperature peak occurring further away from the wellbore. Nevertheless, the low injection rate cases caused the heat to propagate to a longer distance orthogonal to the hydraulic fracture (i.e., more heat was lost to the surrounding porous medium). This is why the area under the curve is not similar in Figure 14b.

#### 4.2.3. TC Concentrations

As shown in Equation (11), the reaction rate depends on the TC concentration. The concentration range usually applied in lab experiments and field practice is between 1:1 and 3:3 mol/L. Figure 15a shows that the lower the TC concentration, the higher the penetration distance. This is because the lower the concentration, the slower the reaction rate. Conversely, higher concentrations result in larger amounts of released heat. This raises the temperature and further increases the reaction rate. The temperature profile strongly depends on the concentration value, as Figure 15b indicates. For instance, 600 °F was reached by applying 3:3 mol/L, while only 280 °F was reached with 1:1 mol/L. This is consistent with the field outcomes, as higher temperatures were recorded when the TC concentrations were greater. Additionally, the temperature peak was closer to the wellbore when the concentration was greater, as the figure shows. This is due to the fast reaction associated with the high concentration scenarios resulting in the heat being released faster. However, the location at which the temperature levels is similar because the fracture size and fluids advection are similar.

#### 4.2.4. Treatment Volume

Different treatment volumes were tested for fixed TC concentrations and injection rates. This was done by increasing the treatment time for each instance. Increasing the treatment volume did not significantly impact the penetration distance of the TCs along the fracture, as Figure 16a shows. Additionally, the treatment volume did not impact the peak temperature value along the fracture length. Nevertheless, it caused heat to propagate a greater distance inside the fracture, as Figure 16b shows. This is because a larger fracture was created, which means that the fluids traveled a longer distance before being lost to the formation. Moreover, the heat propagated a longer distance within the formation orthogonal to the fracture as the treatment volume increased.

#### 4.2.5. TCs Triggered by HCl Acid

Tight carbonate reservoirs are commonly stimulated by acid fracturing. Recently, TC fracture fluids triggered by HCl were implemented in several field trials. The TCs were used to increase the fracture complexity by generating high-pressure pulses, while the acid was used to create the differential etching needed to sustain fracture conductivity after shut-in. The TCs, however, generated high temperatures that significantly impacted the acid’s reactivity with the rock minerals. The model assumed that the TCs and gelled HCl were injected simultaneously, and the reaction took place inside the formation. Figure 17a shows a comparison of the TC-only case to the TCs with HCl case. The figure shows that the acid caused the penetration distance of the TCs to be reduced significantly. This is because the acid reacted with the rock minerals, generating tremendous heat that sped up the TC reaction. Figure 17b shows that the heat generated with the acid was greater, causing the temperature profile to be significantly higher. Acid sped up the reaction of the TCs, causing the peak temperature to occur closer to the wellbore. Notice that the 1000 °F temperature still occurred a couple of feet away from the wellbore.

Figure 18 shows a comparison of the final acid concentration profiles of when only acid was injected and when TCs were injected with the acid. Figure 16a indicates that the acid was consumed within 50 ft when the TCs were injected. Conversely, the acid penetrated to around 150 ft within the fracture when no TCs were used. The short acid penetration length was caused by the extremely high temperature generated by the TCs. Conversely, the TCs created a spike in acid dissolution near the wellbore, as shown in Figure 19a. The location of the dissolution peak is similar to that of the heat-generated peak for the case of TCs with acid. This is because the acid/rock reaction is a strong function of the temperature profile. This is explained by Figure 19b, where the TCs with acid created a much warmer fracture than when only acid was used. This high temperature increased both the diffusion coefficient of the acid and the reaction kinetics of the rock. Acid injection usually causes the temperature to increase slightly, but the synergetic effect of triggering acid with TCs caused the temperature to reach to significantly higher levels. The TCs reduced the acid penetration significantly, but the dissolution magnitude was much higher.

## 5. Conclusions

A coupled TC-reactive transport model for hydraulic fractures was developed in this study. This model is an engineering tool that should be used to optimize TC treatments. The following points summarize the outcomes of this study:1- The treatment design parameters, such as the injection rate, TCs concentration, treatment volume, and TC fluids temperature, impact both the TCs’ penetration distance and heat propagation profile.2- The TC concentration is the most significant in controlling the temperature response and penetration distance of TCs. For instance, the temperature peak could be increased from 280 to 600 °F by increasing the concentration from 1 to 3 mol/L. Nevertheless, the penetration distance would be reduced significantly.3- The injection rate is a significant parameter that could be used to control the TCs penetration distance where higher rates result in higher penetration.4- Most of the cases showed that the TCs are consumed within a short distance from the wellbore; nevertheless, the heat of reaction propagated much longer within the fracture.5- The treatment volume of TCs did not have a significant impact on the temperature magnitude but impacted the fracture volume.6- Acid could be used to trigger the TC reaction, especially in acid fracturing operations. It results in a higher temperature response and higher fracture width and dissolution. The negative impact, however, would be lowering the acid penetration distance.

## Figures and Tables

**Figure 1 molecules-25-04179-f001:**
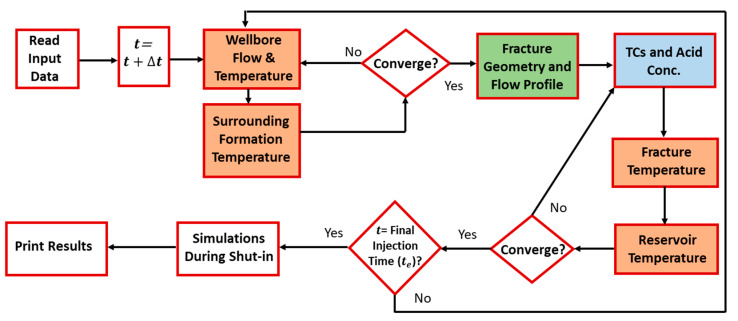
Model algorithm of the fluids’ reactive transport.

**Figure 2 molecules-25-04179-f002:**
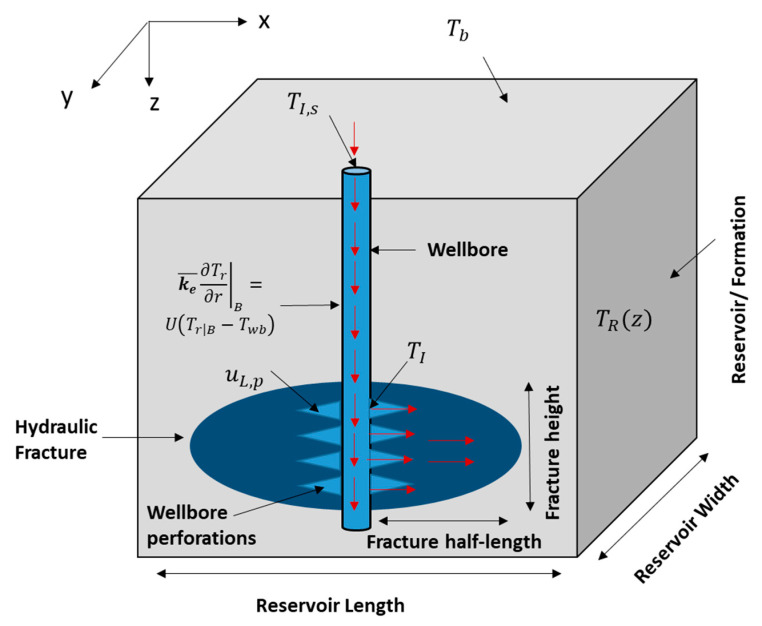
Schematic of the domain (side view) showing the wellbore, hydraulic fracture, and the reservoir.

**Figure 3 molecules-25-04179-f003:**
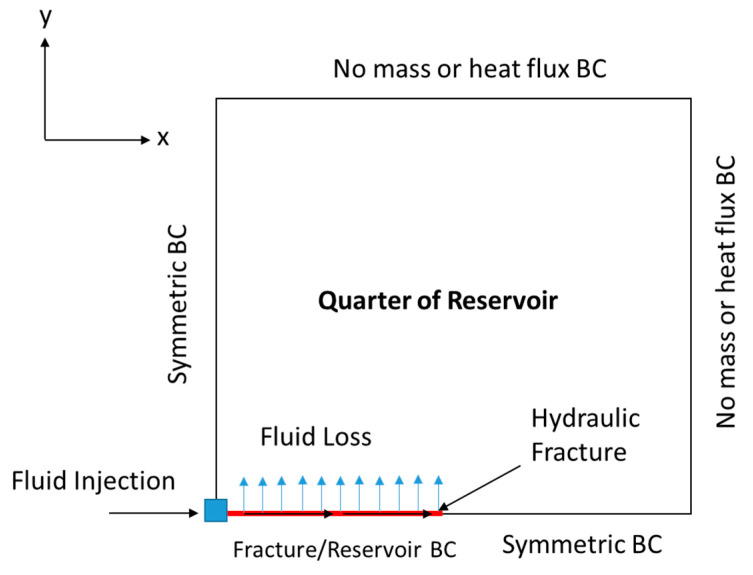
Schematic (top view) showing one quarter of the reservoir domain, including the location of the wellbore and hydraulic fracture.

**Figure 4 molecules-25-04179-f004:**
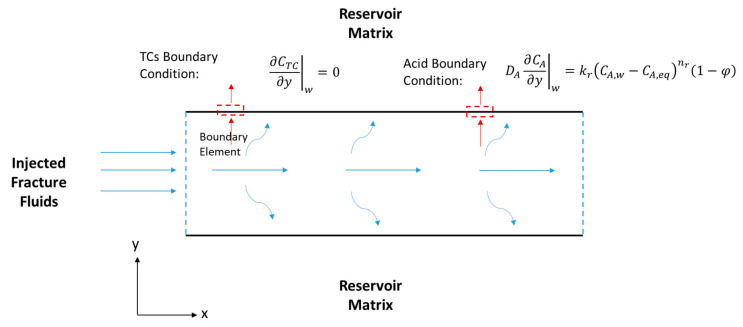
Schematic of a hydraulic fracture along the fracture length (x) and width (y), showing the acid and TCs’ BCs.

**Figure 5 molecules-25-04179-f005:**
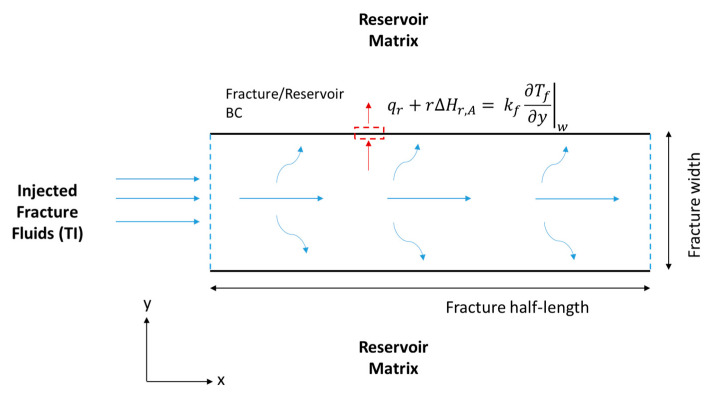
Schematic of the hydraulic fracture along the fracture length (x) and width (y), showing the energy balance BCs.

**Figure 6 molecules-25-04179-f006:**
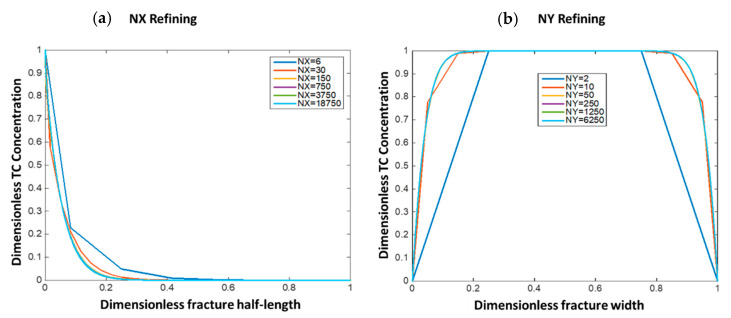
The impact of grid refining on solution convergence for both (**a**) x-direction and (**b**) y-direction.

**Figure 7 molecules-25-04179-f007:**
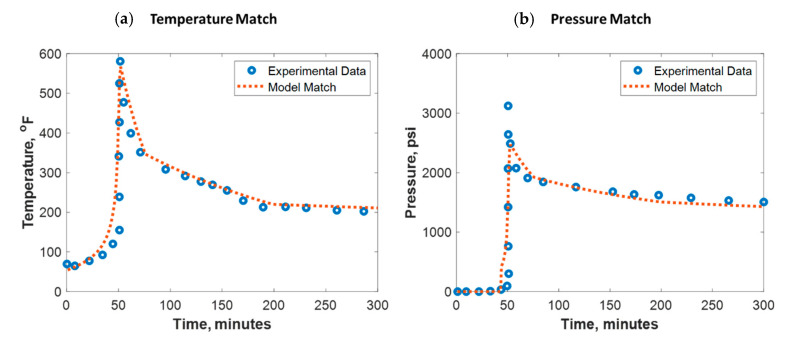
(**a**) Temperature and (**b**) pressure match against experimental data conducted in an autoclave reactor.

**Figure 8 molecules-25-04179-f008:**
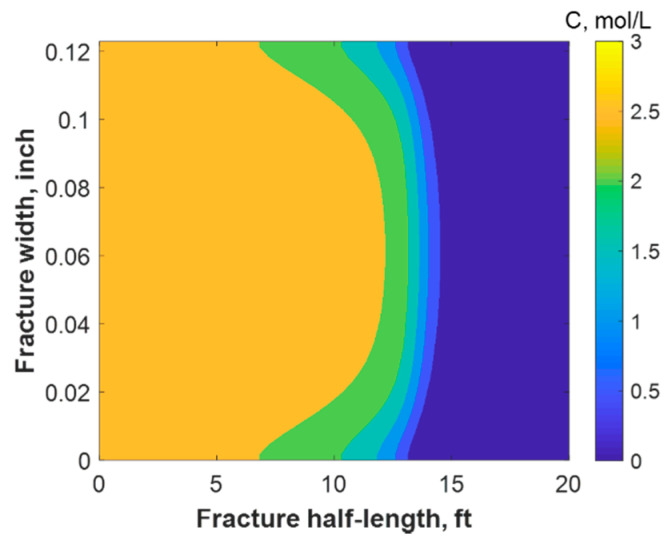
2-D concentration profile of the TCs inside the fracture at the final time step.

**Figure 9 molecules-25-04179-f009:**
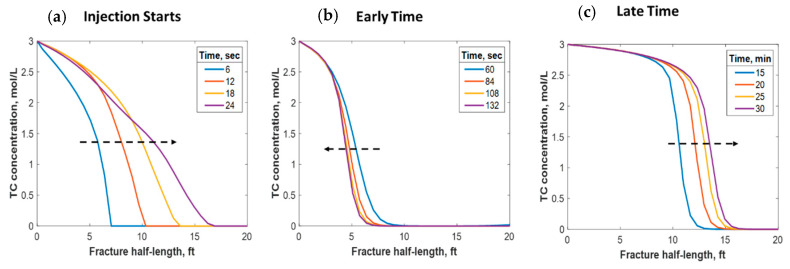
1-D concentration profile of the TCs at (**a**) initial, (**b**) early, and (**c**) late injection times.

**Figure 10 molecules-25-04179-f010:**
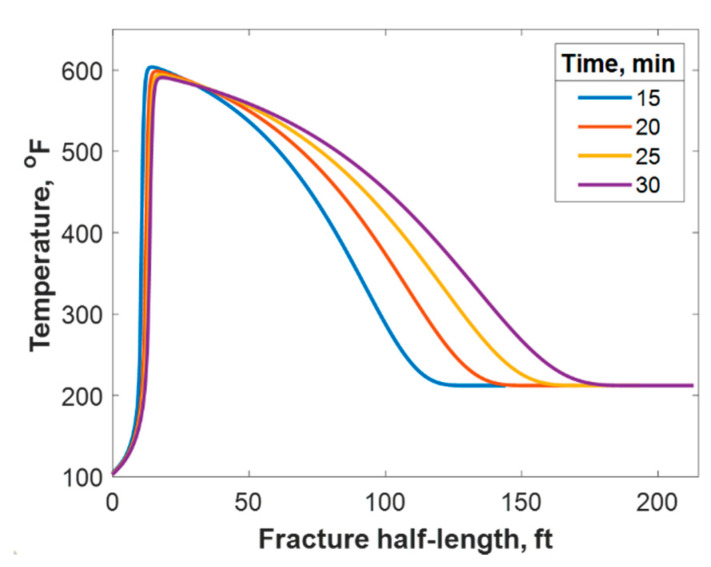
Temperature profile along the fracture half-length at different injection times.

**Figure 11 molecules-25-04179-f011:**
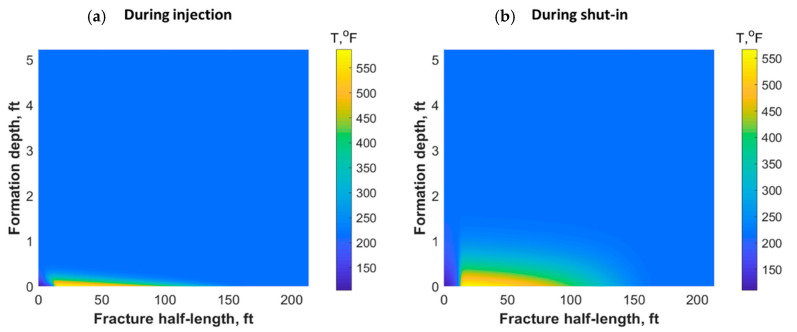
Reservoir temperature profile during (**a**) 30 min of TC injection and (**b**) after 7 h of shut-in.

**Figure 12 molecules-25-04179-f012:**
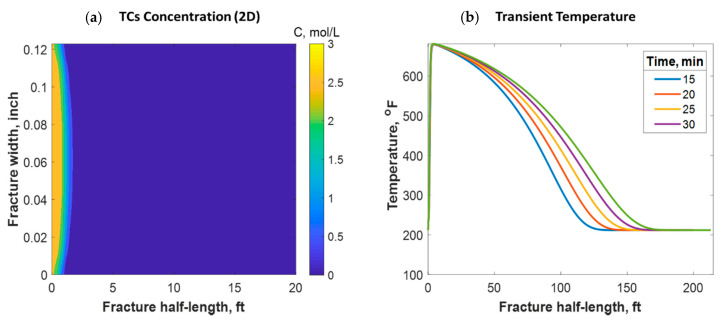
(**a**) 2-D TC concentration profile at the final injection time when the fluids were injected at 212 °F; (**b**) temperature profiles along the fracture at different times.

**Figure 13 molecules-25-04179-f013:**
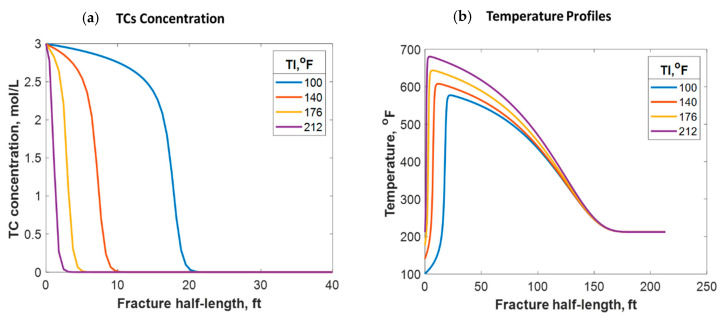
(**a**) TC concentrations and (**b**) fluid temperature profiles along the fracture half-length at different injection temperatures.

**Figure 14 molecules-25-04179-f014:**
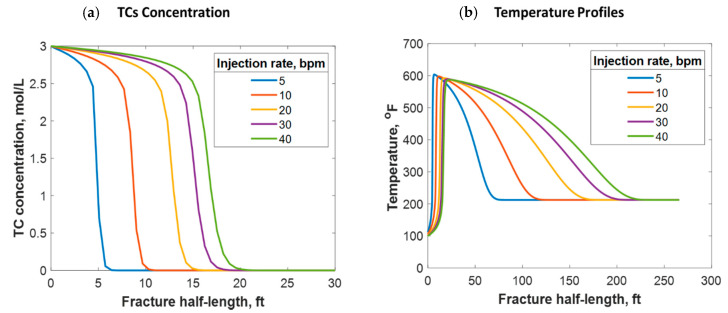
(**a**) TC concentrations and (**b**) fluid temperature profiles along the fracture half-length at different injection rates.

**Figure 15 molecules-25-04179-f015:**
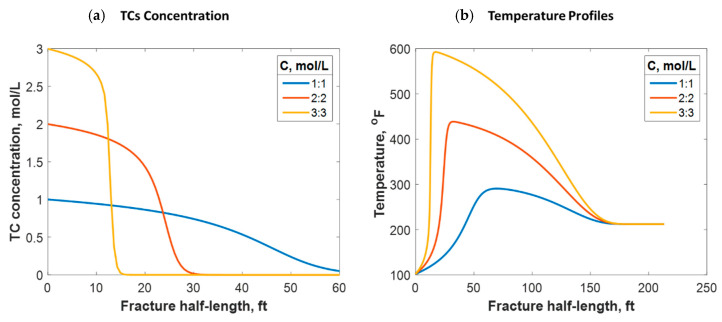
(**a**) TC concentrations and (**b**) fluid temperature profiles along the fracture half-length at different TC concentration values.

**Figure 16 molecules-25-04179-f016:**
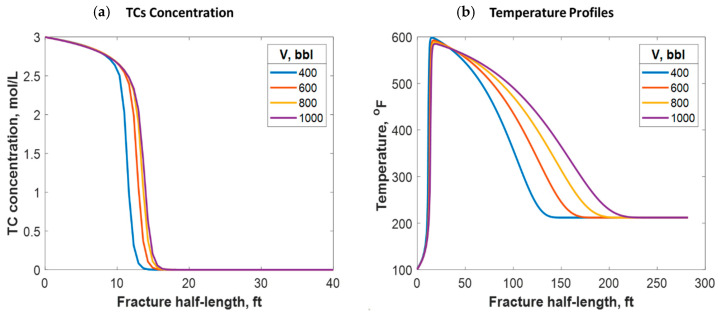
(**a**) TC concentrations and (**b**) fluid temperature profiles along the fracture half-length at different treatment volumes.

**Figure 17 molecules-25-04179-f017:**
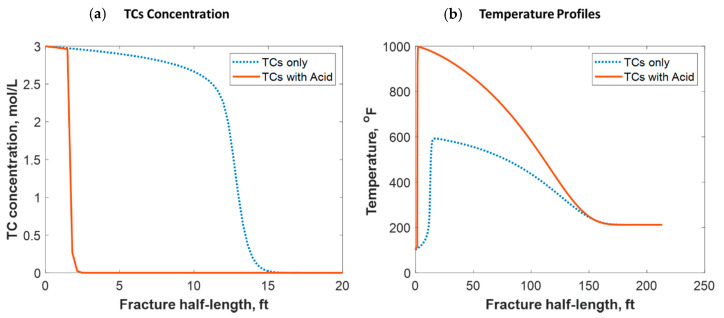
Comparison of (**a**) TC concentrations and (**b**) fluid temperature profiles along the fracture half-length when only TCs were injected versus TCs with acid.

**Figure 18 molecules-25-04179-f018:**
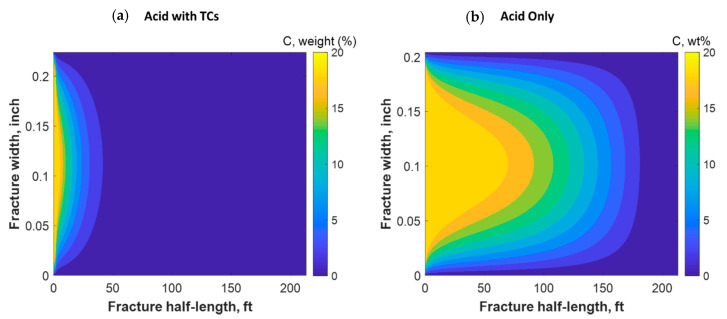
Comparison of acid concentration profiles when (**a**) acid was injected with TCs versus the (**b**) acid-only injection.

**Figure 19 molecules-25-04179-f019:**
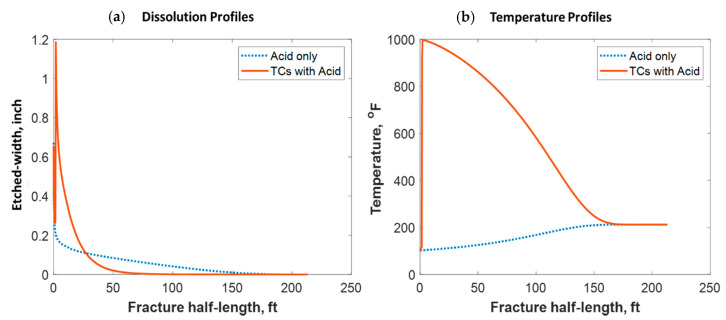
Comparison of the dissolution and temperature profiles when (**a**) acid with TCs was injected versus the (**b**) acid-only injection.

**Table 1 molecules-25-04179-t001:** Input data for the simulations [27,28].

Input Data	SI Unit	Field Unit
Wellbore Property		
Inner casing radius, $r_{1}$	0.0628 m	2.475 inch
Overall heat transfer coefficient, $U_{t}$	0.1 KJ/(s·m^2^ °C)	0.0048 Btu/(hr·ft^2^ °F)
Ambient temperature, $T_{b}$	25 °C	77 °F
Reservoir/Formation Property		
Reservoir pressure,${\;P}_{r}$	2.0684 × 10^7^ pa	3000 psi
Formation fluid density,${\;\rho }_{f}$	850 Kg/m^3^	54 lbm/ft^3^
Formation fluid viscosity, $\mu_{f}$	0.0008 Kg/(m·s)	0.8 cp
Tal compressibility, $c_{t}$	2.26 × 10^−9^ pa^−1^	1.56 × 10^−5^ psi^−1^
Reservoir temperature, $T_{R}$	100 °C	212 °F
Formation rock density,${\;\rho }_{\mathrm{ma}}$	2700 Kg/m^3^	168.5 lb_m_/ft^3^
Formation thickness, $h_{\mathrm{pay}}$	100 ft	30.5 m
Formation specific heat capacity, $c_{\mathrm{ma}}$	0.879 KJ/(Kg °C)	0.2099 Btu/(lb. °F)
Formation thermal conductivity, $k_{\mathrm{ma}}$	1.57 × 10^−3^ KJ/(s·m °C)	0.907 Btu/(hr·ft °F)
Young’s modulus, E	3.1 × 10^9^ pa	4.5 × 10^6^ psi
Closure stress, σ	3.45 × 10^7^ pa	5000 psi
Poisson’s ratio, v	0.25	
Fluid Property		
Density, ρ	1070 Kg/m^3^	66.8 lb_m_/ft^3^
Opening time distribution factor, κ	1.5	
Acid initial concentration,${\;C}_{i}$	0.20 mass HCl/mass solution	
Diffusion coefficient, D	5 × 10^−5^ cm^2^/s	
Fluid specific heat capacity,${\;c}_{p}$	4.13 KJ/(Kg °C)	0.964 Btu/(lb_m_ °F)
Power law exponent, n	0.9	
Consistency index, K	0.002 lbf.s^n^/ft^2^	
Fluid loss coefficient, $C_{L}$	0.004 ft/min^0.5^	
Fluid thermal conductivity, $k_{f}\;$	6 × 10^−4^ KJ/(s·m °C)	0.347 Btu/(hr·ft °F)
Fluid temperature at injection, $T_{I}$	35 °C	95 °F

**Table 2 molecules-25-04179-t002:** Reaction kinetics constants and heat of reaction for acid and TC [31,33].

Reaction	$n_{r}\;$	$k_{r}^{0}\;$	$\frac{\Delta E}{R}\;\left(K\right)\;$	$\Delta H_{r}\;\left(\frac{KJ}{mol\;}\right)\;$
$2HCl+CaCO_{3}$	0.63	7.314 × 10^7^ $\left[\;\frac{kg\;moles\;HCl}{m^{2}\cdot s\cdot \left(\frac{kgmoles\;HCl}{m^{3}\;acid\;solution}\right){\;}^{n_{r}}}\right]$	7.55 × 10^3^	7.5
$NH_{4}Cl+NaNO_{2}$	1	9.99 × 10^3^ 1/s	4.58 × 10^3^	368

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
