# Peer review of "Mass and Heat Transfer of Thermochemical Fluids in a Fractured Porous Medium"

_molecules, 2020, doi:10.3390/molecules25184179_

Round 1
Reviewer 1 Report
This manuscript studies the penetration length of injected thermochemical fluids and corresponding generated heat for improving the treatment of hydraulic fracturing. Pure numerical approach is employed for this study. It is a important study for the oil recovery, especially for the production of shale gas and oil. I feel this study will also interest the reader of Molecules. However, there are some major and minor concerns of this manuscript before I can endorse its publication on Molecules.
Major Concerns:
- This is a pure numerical approach for a case-like study. There is no experimental results or field data to validate the numerical study. This is a big concern for this study.
- What is the solver used for this study? The credibility of the solver is important for the numerical results.
- It is apparent that the reaction can generate some gases. How will these gases or potential multiphase flow affect the transport of the mass and heat? The manuscript has no discussion about it.
Minor Concerns:
- Why do authors choose to set the penetration distance as the location inside the fracture where the TCs concentration is zero or the temperature drops to the initial reservoir temperature? I think most of transport studies choose use the half of the maximum concentration as a reference due to the tailing or fingering effects. It is expected the authors have a discussion about it.
- Do the authors assume all the chemicals are well mixed for reaction? Actually, the well mixing assumption for reactive transport can cause a big deviation from the experimental observations.
- What is the triggering temperature for the TCs reaction? I cannot find it in the manuscript. Will the reaction rate change with temperature and pressure? I hope the authors can clarify it in the revised version.
- The authors state "The model is dynamic, in that the fracture propagates while the fluids containing TCs are being injected" at Line 281-282. However, in the following method description, I didn't see the algorithms for simulating the fracture propagation. It should be clarified.
Author Response
I would like to thank the editor and reviewers for their thorough feedback. We addressed all the points raised by reviewers and it significantly improved the paper quality. Please find below our response in point by point bases. You will find our response in red while the modification in the manuscript you fill find it highlighted in yellow.
Reviewer 1
This manuscript studies the penetration length of injected thermochemical fluids and corresponding generated heat for improving the treatment of hydraulic fracturing. Pure numerical approach is employed for this study. It is a important study for the oil recovery, especially for the production of shale gas and oil. I feel this study will also interest the reader of Molecules. However, there are some major and minor concerns of this manuscript before I can endorse its publication on Molecules.
Major Concerns:
- This is a pure numerical approach for a case-like study. There is no experimental results or field data to validate the numerical study. This is a big concern for this study.
Response: This is a valuable comment. We added a section called model verification where we verified our model against published experimental data. A reasonable match was obtained between the model and experiments.
- What is the solver used for this study? The credibility of the solver is important for the numerical results.
Response: We developed the model in house using MATLAB. No other commercial software was used. The following was added to the methodology section:
The model was developed in-house using MATLAB
- It is apparent that the reaction can generate some gases. How will these gases or potential multiphase flow affect the transport of the mass and heat? The manuscript has no discussion about it.
Response: Multiphase flow was not considered in this study as the model will be very complex. We added a section named model limitations. The following sentence was added to that section:
Modeling TCs reactive transport is a complex phenomenon as the reaction results in multiphase flow and possible turbulence. The developed model assumes that the TCs are incompressible (even after reactions) and the flow regime to be laminar. The component mass transport accounts for the reactants concentration but ignore the products of reaction such as CO2 and N2.
Minor Concerns:
- Why do authors choose to set the penetration distance as the location inside the fracture where the TCs concentration is zero or the temperature drops to the initial reservoir temperature? I think most of transport studies choose use the half of the maximum concentration as a reference due to the tailing or fingering effects. It is expected the authors have a discussion about it.
Response: I have seen some studies that they use 10% of the initial concentration as penetration distance (90% consumption) especially for acid fracturing. We believe in this study; it won’t make much difference due to sharp front. Also, pure TCs might not result in viscous fingering effect. We added the following sentence for explanations:
Other studies prefer to use a percentage of the initial chemical concentration for penetration distance calculation. Thermochemical reaction usually results in sharp concentration front which makes indifferent to use zero concentration or other slightly higher concentration.
- Do the authors assume all the chemicals are well mixed for reaction? Actually, the well mixing assumption for reactive transport can cause a big deviation from the experimental observations.
Response: This is a good point. We assumed that the chemicals were co-injected which is common in field practices resulting in well mixed mixture. It might not be always the case, however, as field operations are far from ideal. We added the following sentence to the model limitation section.
Also, the thermochemical fluids were assumed to be well mixed while flowing in the fracture which is reasonable if they were co-injected at the wellbore perforations.
- What is the triggering temperature for the TCs reaction? I cannot find it in the manuscript. Will the reaction rate change with temperature and pressure? I hope the authors can clarify it in the revised version.
Response: We assumed that once it hits the formation, the reaction takes place. The fluids in this study reach the formation at 102 oF according. Hence the temperature is triggered at 102 oF. We added the following sentence to section 3.
In this study, the reaction is assumed to be triggered at 102 oF. It should be noted, however, that the triggering temperature could be altered according to the specification of a certain field
The reaction depends strong on temperature but has no dependence on pressure. We added the following comments
Notice that both the thermochemical fluids and acid reaction rates depend strongly on temperature according to Eq. 18 but has no dependence on pressure
- The authors state "The model is dynamic, in that the fracture propagates while the fluids containing TCs are being injected" at Line 281-282. However, in the following method description, I didn't see the algorithms for simulating the fracture propagation. It should be clarified.
Response: we used Equations 24 and 25 to estimate the fracture half-length and width at each times step. These equations according to the fluid loss coefficient can be used to track fracture propagation based on fracture material balance. In other words, based on material balance, the fracture volume will be determined at each time step.
Reviewer 2 Report
A numerical study on heat and mass transfer of thermochemical fluids in a fractured porous medium is reported in the present paper. The manuscript is well-structured and the quality of figures are acceptable. There are however minor deficiencies that should be addressed before considering this work for publication.
- The major findings of the present work should be highlighted in the abstract.
- The results of grid independence test should be reported in the paper.
- The validity and reliability of numerical simulations should be verified by making comparisons with experimental and theoretical results published in the literature.
- Conclusions are a summary of the results. This section should be revised.
Author Response
I would like to thank the editor and reviewers for their thorough feedback. We addressed all the points raised by reviewers and it significantly improved the paper quality. Please find below our response in point by point bases. You will find our response in red while the modification in the manuscript you fill find it highlighted in yellow.
Reviewer 2
A numerical study on heat and mass transfer of thermochemical fluids in a fractured porous medium is reported in the present paper. The manuscript is well-structured and the quality of figures are acceptable. There are however minor deficiencies that should be addressed before considering this work for publication.
- The major findings of the present work should be highlighted in the abstract.
Response: We agree with the reviewer. We added the following sentences to the abstract
Among other design parameters, the thermochemical fluid concentration is the most significant in controlling the penetration length, temperature, and pressure response. Also, Acid can be used to trigger the reaction but results in shorter penetration length and higher temperature response.
- The results of grid independence test should be reported in the paper.
Response: We added the following section
2.5 Solution Grid Independence
This section is showing the optimum grid size selection for the numerical solutions in this study. This was done by refining the grid blocks in both the fracture length (NX) and fracture width (NY) directions and observe the solution error reduction as compared to the most refined solution (NX=18,750 and NY= 6,250). Figure 6 shows the solutions in dimensionless format at different grid sizes. Figure 6a shows that when NX=150, the solution becomes almost similar to the most refined solution and results in small root mean square error (RMSE =1x10-3). Figure 6b shows that when NY= 50, the results become similar to most refined solution with RMSE = 10-4. In this study, NX=250 and NY= 80 were selected in the simulations to lower the solution error without high computational cost.
- The validity and reliability of numerical simulations should be verified by making comparisons with experimental and theoretical results published in the literature.
Response: We added a section called model verification where we verified our model against published experimental data. A reasonable match was obtained between the model and experiments.
- Conclusions are a summary of the results. This section should be revised.
Response: We re-wrote that section in summary form as suggested by the reviewer.

Reviewer 3 Report
This is a great research paper with extensive modeling work. However there is no validation of the simulation results nor much theoretical data to support the outputs and conclusions of the simulations. I recommend that you provide some form of validation for all the parameters that were studied. It would be great if actual field data could be used and simulations predict closely the actual field findings. I have some specific comments that are shown below:
1) Are the data used in Table 1 and 2 are hypothetical or actual field data? If hypothetical, they need to be justified. If field data, should have reference.
2) Table 1 shows HCl acid behaves as powe-law fluid, however my understanding is that pure HCl behaves as a Newtonian fluid. Please provide reference.
3) Are data in Table 2 measured? If not, why not measured?
4) Figure 3 does not support above statements as temperature seems to have increased with time up to 15 minutes. Even after 15 minutes, temperature decrease with time seems to be very insignificant for the first 15 ft to explain the difference in outcome. Should show temperature profile for start injection times and early times in Figure 3 to support Figure 2 output.
5) What is the reason for this specific distance for max temperature? Any physics behind this.
6) Why is the concentration profile(figure 1) a sharp quick decline (like a shock front) with no distribution?
7) IS the reaction of TC mass transfer or surface reaction controlled? How was this determined?
8) You show temperature increases to 1000 oF with HCl. Can you show data from other studies showing such a large temp increase with HCl?
Author Response
I would like to thank the editor and reviewers for their thorough feedback. We addressed all the points raised by reviewers and it significantly improved the paper quality. Please find below our response in point by point bases. You will find our response in red while the modification in the manuscript you fill find it highlighted in yellow.
Reviewer 3
This is a great research paper with extensive modeling work. However, there is no validation of the simulation results nor much theoretical data to support the outputs and conclusions of the simulations. I recommend that you provide some form of validation for all the parameters that were studied. It would be great if actual field data could be used and simulations predict closely the actual field findings. I have some specific comments that are shown below:
Response: Thanks for the encouraging comments. Field data of TCs treatment are rare. However, we added a section called model verification where we verified our model against published experimental data. A reasonable match was obtained between the model and experiments.
1) Are the data used in Table 1 and 2 are hypothetical or actual field data? If hypothetical, they need to be justified. If field data, should have reference.
Response: The data from Table 1 are based on previous publications of typical field data. The data from Table 2 are based on lab studies of reaction kinetics. I added the references for both tables (please look at the table captions). I also added the following sentences:
Tables 1 and 2 show the data used for the simulations employed in this study. The data in Table 1 are based on typical wellbore, reservoir, and fluids properties that are encountered in carbonate reservoirs. The data in Table 2 are based on lab reaction kinetics measurements.
2) Table 1 shows HCl acid behaves as power-law fluid, however my understanding is that pure HCl behaves as a Newtonian fluid. Please provide reference.
Response: This is true. Pure HCl is Newtonian fluid. However, usually emulsified or gelled acids are injected which behave like power law fluids. I added the references for Table 1. I modified one of the sentences in section 3.2.5 to include the word gelled as follows:
The model assumed that the TCs and gelled HCl were injected simultaneously
3) Are data in Table 2 measured? If not, why not measured?
Response: They are lab measured data from two different sources. Please look at reference 31 for reaction kinetics of TCs and reference 33 for reaction kinetics of acid. I already added the references to the Table caption.
4) Figure 3 does not support above statements as temperature seems to have increased with time up to 15 minutes. Even after 15 minutes, temperature decrease with time seems to be very insignificant for the first 15 ft to explain the difference in outcome. Should show temperature profile for start injection times and early times in Figure 3 to support Figure 2 output.
Response: The temperature does not decrease with time, only the penetration distance of TC does on early time. So, the temperature will keep increasing and later will stay constant within the 15 ft. However, the concentration behavior will change in early times.
5) What is the reason for this specific distance for max temperature? Any physics behind this.
Response: This is associated with fluid convection and the reaction being fastest away from the wellbore. Similar outcomes are observed for HCl acid in previous studies where the max temperature occurs away from the inlet due to heat convection. I added the following sentences with two references to support the outcomes:
The initial sharp increase in temperature is caused by the fast exothermic reaction which was assumed to be initiated in the reservoir. The temperature peak is associated with location at which the thermochemical fluids are completely consumed. Then, the temperature decreases due to the heat loss to the colder reservoir as fluids are flowing in the fracture. At a certain point, fracture fluids temperature reaches a plateau which is the initial reservoir temperature as illustrated in Figure 8. Many fundamental studies concluded that the temperature peak occurs away from the wellbore in reactive transport problems [34, 35].
6) Why is the concentration profile (figure 1) a sharp quick decline (like a shock front) with no distribution?
Response: I believe this is due to the type of reaction. TCs at a certain temperature reacts instantaneously and are consumed fast. Also, diffusion is not important and the problem is pure convection and reaction. These type of problems usually result in sharp front. I added the following sentence regarding ignoring diffusion.
The diffusion term can be ignored as it has no noticeable impact on the solution
7) IS the reaction of TC mass transfer or surface reaction controlled? How was this determined?
Response: The mass transfer or surface reaction phenomena occurs for heterogeneous reactions. The TC reaction is homogenous (occurs in the liquid phase), so it is neither one. For acid reaction with limestone, it is always mass transfer as the surface reaction is infinite and slowest step (mass transfer) determines the overall reaction.
8) You show temperature increases to 1000 oF with HCl. Can you show data from other studies showing such a large temp increase with HCl?
Response: HCl by itself will result only in few degree increase in temperature as many studies indicated. However, the combination of HCl and TC is what causes the temperature to reach 1000 oF. I have not seen any public data of field studies of TCs triggered with acid. However, from discussion with our partners in Saudi Aramco, the temperature could reach up to 600 oF near the wellbore. The 1000 oF reported in this study could not be sensed in the wellbore as it occurs couple of feet away but a medium temperature such as 600 oF will be sensed using temperature sensing tools. The following sentence was added:
Acid injection by itself usually causes the temperature to increase slightly but the synergetic effect of triggering acid with TCs caused the temperature to reach to significantly higher levels.
Reviewer 4 Report
The current proposed paper is presenting a model of limestone (CaCO3) using a thermochemical fluid made of a solution Ammonium Chloride (NH3Cl) combined with Sodium Nitrite (NaNO2). Simulation results (such as the profiles of temperatures, concentrations and fracture widths) are presented including comparison with conventional use of Hydrochloric acid solution (HCl). The application is useful and the present work has some novelty. However the current draft did not follow a rigorous methodology of writing a research paper and is far from meeting the standard of scientific publication. More simple structure needs to be adopted:
- Introduction
- Modelling Methodology
- Simulation Results and Analysis
- Conclusion
More importantly, the description of the modelling methodology must come after the Introduction section. It must also show with clarity the following:
- Description of the system you want to model.
- Relevant equations (mainly mass, heat and momentum balances equations) underpinning all processes occurring including justifications and any assumption as possible.
- Operation conditions of simulation.
There are also many other aspects including English grammar that need substantial improvements. Perhaps the authors could consider a prove reading of the draft before submission.
See attached file for more detailed comments and suggestions.

Author Response
I would like to thank the editor and reviewers for their thorough feedback. We addressed all the points raised by reviewers and it significantly improved the paper quality. Please find below our response in point by point bases. You will find our response in red while the modification in the manuscript you fill find it highlighted in yellow.
Reviewer 4
The current proposed paper is presenting a model of limestone (CaCO3) using a thermochemical fluid made of a solution Ammonium Chloride (NH3Cl) combined with Sodium Nitrite (NaNO2). Simulation results (such as the profiles of temperatures, concentrations and fracture widths) are presented including comparison with conventional use of Hydrochloric acid solution (HCl). The application is useful and the present work has some novelty. However the current draft did not follow a rigorous methodology of writing a research paper and is far from meeting the standard of scientific publication. More simple structure needs to be adopted:
- Introduction
- Modelling Methodology
- Simulation Results and Analysis
- Conclusion
More importantly, the description of the modelling methodology must come after the Introduction section.
Response: Thanks for the suggestions. The research paper was restructured as instructed by the reviewer
It must also show with clarity the following:
- Description of the system you want to model.
Response: we modified the schematic figures (Figures 2-5) to make the system clear. I also added the following sentences to describe the system more:
The modeled system contains subterranean tight formation, a wellbore drilled in the middle of the formation, and a fracture extending from the wellbore.
The fluids are injected to the wellbore where they travel down the hole until reaching the wellbore perforations. It creates the two-wing fracture where the pressurized fluids propagate the hydraulic fracture. The red arrows show the directions of the fluids flow in the domain.
Figure 3 shows the two dimensional (2D) schematic of the domain by taking a slice in the z direction (top view).
- Relevant equations (mainly mass, heat and momentum balances equations) underpinning all processes occurring including justifications and any assumption as possible.
- Operation conditions of simulation.
Response: We added a section to describe the model limitations and operations conditions:
Modeling TCs reactive transport is a complex phenomenon as the reaction results in multiphase flow and possible turbulence. The developed model assumes that the TCs are incompressible (even after reactions) and the flow regime to be laminar. The component mass transport accounts for the reactants concentration but ignore the products of reaction such as CO2 and N2. The complex fracture network created by the TCs reaction is also ignored and planar fracture is assumed. Also, the thermochemical fluids were assumed to be well mixed while flowing in the fracture which is reasonable if they were co-injected at the wellbore perforations.
There are also many other aspects including English grammar that need substantial improvements. Perhaps the authors could consider a prove reading of the draft before submission.
Response: We proof read the full manuscript and the changes are highlighted through the manuscript. We also did proof reading before submission with professional copy/editor
Other comments in the attached PDF by reviewer 4 is below
-The abstract must include relevant numerical data.
Response: We added relevant data to the modified abstract
-How do you define fracture efficiency?
Response: In this case, efficiency meant complexity of the fracture. We added the word complexity
-For the conclusion than the introduction.
Response: we moved the sentence to the conclusion
-This table is not part of simulation results but could be included in section namely “Methodology”
Response: we moved the Table to the methodology
-This table must be presented fully in the same page.
Response: we presented the tables in the same page in the new revision
The fracture is underpinned by the double of both pressure and temperature increases due to the chemical reaction. It means that temperature profiles as as pressure profiles must presented.
Response: The paper will be longer if we present all pressure profiles along with concentration and temperature. However, we included the pressure profile in model verification section in Figure 7
-why this peak? why this settling temperature?
Response: We added the following sentences to explain
The initial sharp increase in temperature is caused by the fast exothermic reaction which was assumed to be initiated in the reservoir. The temperature peak is associated with location at which the thermochemical fluids are completely consumed. Then, the temperature decreases due to the heat loss to the colder reservoir as fluids are flowing in the fracture. At a certain point, fracture fluids temperature reaches a plateau which is the initial reservoir temperature as illustrated in Figure 8. Many fundamental studies concluded that the temperature peak occurs away from the wellbore in reactive transport problems [34, 35].
-Definition of terms must absolutely include the associated units using Standard International Units
(Metric and not Imperial)
Response: We added the units for all parameters in the nomenclature both Metric (SI) and field unit. We added field units are it is important for the reader of this work. Petroleum engineers are used to work with field units.
-Figure modifications
Response: Thanks for comments on the figures. We improved the quality of these figures by clarifying the items in the figures.
Round 2
Reviewer 1 Report
The manuscript is well improved after revision.
Author Response
Thanks for you encouraging comments.

Reviewer 3 Report
You have replied to all the comments. I am still concerned about the validation of the model as there was significant need to adjust parameters to match the experimental results.
Author Response
I agree that the range of the overall heat transfer coefficient is large (from 0.01 to 0.4 KJ/(s.m2.oC)).
However, this is one significant unknow that changes significantly due to gas generation during the experiment. So, assuming other thermal and reaction properties are accurate (obtained from literature), only U can be used to match the temperature.

Reviewer 4 Report
Previous assessment of this paper has provided substantial and detailed comments aimed to improve both substance and presentation quality of the initial draft. Although the authors have made few changes accordingly, the current revised version is not up to standard for publication since there are still many relevant issues that are not addressed or are not addressed appropriately.
- Previous review has made grammar corrections fully on “Introduction” as example and it was down to authors to improve further the remaining draft including using proof reading services if necessary. The current draft has no evidence that this was considered.
- Each relevant equation (mass, heat or momentum) underpinning a process occurring must include justifications and any assumption if there is. The definition each term of the equation must the associated SI unit. The current draft has limited evidence that this was considered. Providing a sentence stated some assumptions at the end of the section is not good enough. It must not be down to the reader to work out which equation it is referring to and the rationale behind it.
- Previous review has suggested structural changes to the initial draft and the authors have done so. However they have no or little attention to the text coherence while swapping the two sections (‘Modelling’ and ‘Simulation results’). More importantly there are additional and useful sub-sections (‘Model Limitations’, ‘Solution Grid Independence’ and ‘Model Verification’) which mean that the authors must now adopt different approach. Structuring speaking, the following could be adopted: 1) Introduction; 2) Modelling; 3) Model Validation; 4) Model exploitation; 5) Conclusion. Model Validation section must include detailed description of the experimental set up and the experimental results compared to the model predictions (as per ‘Model Verification’).
- There was a request of explanations of peaks and planes on many figures of the Simulation results. The current draft has no evidence that this was considered. Providing a sentence stated a generic explanation is not good enough. For each relevant and individual figure, explanation must be backed up numerical values. Analysis must also include comparison with information from the literature as possible.
- There was a request to review ‘Reference’ section. The current draft has no evidence that this was considered.
- There was a request to review ‘Nomenclature’ section. The current draft has evidence that this is not done properly. See literature for SI Units.
- There is no justification of using Imperial Units throughout the text including on all figures nowadays when Metric Units are adopted as SI Units.
Author Response
Previous assessment of this paper has provided substantial and detailed comments aimed to improve both substance and presentation quality of the initial draft. Although the authors have made few changes accordingly, the current revised version is not up to standard for publication since there are still many relevant issues that are not addressed or are not addressed appropriately.
- Previous review has made grammar corrections fully on “Introduction” as example and it was down to authors to improve further the remaining draft including using proof reading services if necessary. The current draft has no evidence that this was considered.
We appreciate the reviewer help in correcting the grammatical errors. We corrected few words as highlighted in the manuscript in yellow. Also, attached the original document showing that it was revised by professional copy/editor. I think the reviewer main concern is using the past tense instead of the present tense. We consulted with couple of editors and all agree that using either past tense or present tense should be fine. However, they preferred the usage of past tense as much of the research has already occurred.
- Each relevant equation (mass, heat or momentum) underpinning a process occurring must include justifications and any assumption if there is. The definition each term of the equation must the associated SI unit. The current draft has limited evidence that this was considered. Providing a sentence stated some assumptions at the end of the section is not good enough. It must not be down to the reader to work out which equation it is referring to and the rationale behind it.
Thanks for the valuable suggestions. In each equation, I stated the purpose of it by showing what parameter it estimates and in which domain. For example, equation 2 is used to estimate the velocity in the wellbore. I also stated the meaning of each term in that differential equation. The same was applied for all the differential equations provided in the study. Finally, as suggested by the reviewer, I stated the assumptions of each differential equation as highlighted in yellow. The definition of each term in SI units was provided as requested. Boundary conditions were also illustrated.
- Previous review has suggested structural changes to the initial draft and the authors have done so. However they have no or little attention to the text coherence while swapping the two sections (‘Modelling’ and ‘Simulation results’). More importantly there are additional and useful sub-sections (‘Model Limitations’, ‘Solution Grid Independence’ and ‘Model Verification’) which mean that the authors must now adopt different approach. Structuring speaking, the following could be adopted: 1) Introduction; 2) Modelling; 3) Model Validation; 4) Model exploitation; 5) Conclusion. Model Validation section must include detailed description of the experimental set up and the experimental results compared to the model predictions (as per ‘Model Verification’).
We restructured the paper as requested by the reviewer. We added the following for the experiment description.
The reactor is manufactured to handle high pressure and high temperature (HPHT) environment. The reaction is monitored remotely using PC for safety purposes where the temperature and pressure are recorded every 2 seconds. These reactors are usually made of high-quality jacket that is made of stainless steel and inner Teflon champers. Al-Nakhli et al. specified the different reactor volumes used and their pressure and temperature ratings in details [36].
- There was a request of explanations of peaks and planes on many figures of the Simulation results. The current draft has no evidence that this was considered. Providing a sentence stated a generic explanation is not good enough. For each relevant and individual figure, explanation must be backed up numerical values. Analysis must also include comparison with information from the literature as possible.
We tried to explain the peaks for each individual case as highlighted in the manuscript.
I would like to illustrate that we are the first to model thermochemical fluids reaction in reservoir conditions. So, there is no literature to compare with. However, there are publications on reactive transport of HCl acid only that shows the peak of heat occurring away from the wellbore. We already referred to these papers within the text according the last review round.
- There was a request to review ‘Reference’ section. The current draft has no evidence that this was considered.
The citations were fixed as requested.
- There was a request to review ‘Nomenclature’ section. The current draft has evidence that this is not done properly. See literature for SI Units.
I added only SI Units for Nomenclature as requested by the reviewer.
- There is no justification of using Imperial Units throughout the text including on all figures nowadays when Metric Units are adopted as SI Units.
I totally agree with the reviewer as my background is chemical engineering. However, in the petroleum industry, they are still heavily using what they call Field units. The audience of this paper will better understand it this way. For instance, if we mention fracture width or length in m, petroleum engineers cannot relate. However, we added the units in SI in the nomenclature as well as in the methodology.
